# Reduced OPA1, Mitochondrial Fragmentation and Increased Susceptibility to Apoptosis in Granular Corneal Dystrophy Type 2 Corneal Fibroblasts

**DOI:** 10.3390/genes14030566

**Published:** 2023-02-24

**Authors:** Seung-Il Choi, Ga-Hyun Lee, Jong-Hwan Woo, Ikhyun Jun, Eung Kweon Kim

**Affiliations:** 1Corneal Dystrophy Research Institute, Yonsei University College of Medicine, Seoul 03722, Republic of Korea; 2Department of Food Marketing and Safety, Konkuk University, Seoul 05029, Republic of Korea; 3The Institute of Vision Research, Department of Ophthalmology, Yonsei University College of Medicine, Seoul 03722, Republic of Korea; 4Saevit Eye Hospital, Goyang-si 10447, Republic of Korea

**Keywords:** mitochondria, OPA1, NRF1, PGC1α, fission and fusion, corneal fibroblasts, granular corneal dystrophy type 2

## Abstract

The progressive degeneration of granular corneal dystrophy type 2 (GCD2) corneal fibroblasts is associated with altered mitochondrial function, but the underlying mechanisms are incompletely understood. We investigated whether an imbalance of mitochondrial dynamics contributes to mitochondrial dysfunction of GCD2 corneal fibroblasts. Transmission electron microscopy revealed several small, structurally abnormal mitochondria with altered cristae morphology in GCD2 corneal fibroblasts. Confocal microscopy showed enhanced mitochondrial fission and fragmented mitochondrial tubular networks. Western blotting revealed higher levels of MFN1, MFN2, and pDRP1 and decreased levels of OPA1 and FIS1 in GCD2. OPA1 reduction by short hairpin RNA (shRNA) resulted in fragmented mitochondrial tubular networks and increased susceptibility to mitochondrial stress-induced apoptosis. A decrease in the mitochondrial biogenesis-related transcription factors NRF1 and PGC1α was observed, while there was an increase in the mitochondrial membrane proteins TOM20 and TIM23. Additionally, reduced levels of mitochondrial DNA (mtDNA) were exhibited in GCD2 corneal fibroblasts. These observations suggest that altered mitochondrial fission/fusion and biogenesis are the critical molecular mechanisms that cause mitochondrial dysfunction contributing to the degeneration of GCD2 corneal fibroblasts.

## 1. Introduction

Granular corneal dystrophy type 2 (GCD2, also previously called Avellino corneal dystrophy) is an autosomal dominant disorder caused by pathogenic variant p.(Arg124His) in the transforming growth factor-β-induced gene (*TGFBI*) on chromosome 5q31 [1]. Age-dependent progressive accumulation of mutant TGFBI protein (TGFBIp) in the corneal stroma is the hallmark of *TGFBI*-linked corneal dystrophy, including GCD2, and interferes with corneal transparency [2,3,4].

The corneal stroma constitutes approximately 90% of human corneal thickness and consists of an extracellular matrix (ECM) interspersed with quiescent keratocytes [5], also called corneal fibroblasts [5,6]. The keratocytes are responsible for maintaining and repairing the matrix and producing collagens and proteoglycans in the corneal stroma [7,8], which is composed of layers of collagen fibrils coated with different proteoglycans [9]. The pathogenic mutant TGFBIp accumulates between the collagen lamellae of the corneal stroma. Therefore, the function of corneal fibroblasts may play an important role in mutant TGFBIp deposition.

Mitochondria are essential for all eukaryotic cells and play key roles in energy metabolism, cell survival, apoptosis, and cellular signaling [10,11]. Mitochondrial dynamics not only determines their morphology and size but also regulates their distribution and function [12]. Disruptions in the fusion and fission processes can cause several diseases [13], indicating that the maintenance of mitochondrial morphology is critical for normal cell function. Recently, several studies have highlighted the importance of mitochondrial dynamics in cellular physiological and various pathological conditions, showing the change of dynamics dramatically alters mitochondrial morphology [14].

Mitochondrial structural dynamics is controlled and maintained by at least five proteins: optic atrophy 1 (OPA1), mitofusins 1 and mitofusins 2 (MFN1 and MFN2) that promote mitochondrial fusion and fission 1 (FIS1), and dynamin-related protein 1 (DRP1) that are essential for mitochondrial fission [12,15]. Among the fission and fusion proteins, OPA1 is essential for mitochondrial membrane fusion [16,17,18,19,20] and mitochondrial DNA (mtDNA) maintenance [21,22] including mtDNA stability and replication. Moreover, OPA1 oligomerization is involved in mitochondrial cristae structure and protection against apoptosis [23,24,25,26,27]. Several *OPA1* mutations are known to cause autosomal dominant optic atrophy [10,28].

Cells regulate mitochondrial mass through biogenesis [29], which is a complex and multistep processes requiring replication of mtDNA, synthesis of inner and outer mitochondrial membrane, synthesis of mitochondrial-encoded proteins, and import and synthesis of nuclear-encoded mitochondrial proteins. The transcriptional coactivator peroxisome proliferator-activated receptor γ coactivator-1 α (PGC-1α) regulates mitochondrial biogenesis and energy metabolism by interacting with nuclear respiratory factors 1 and 2 (NRF1 and NRF2) [30,31]. These two key transcription factors activate the expression of the mitochondrial transcription factor A (Tfam) that is essential for mtDNA transcription and replication. These factors also regulate the five complexes in the mitochondrial electron transport chain, import of nuclear-encoded mitochondrial proteins, and mtDNA replication and transcription [32].

Pathophysiological studies of GCD2 corneal fibroblasts have revealed mitochondrial oxidative stress [33] and altered mitochondrial function [34], including increased mitochondrial fragmentation [35] in GCD2 corneal fibroblasts. Despite these notable observations, the mechanisms governing this dysfunction are yet to be fully elucidated. Here, we investigate mitochondrial dynamics and biogenesis in GCD2 corneal fibroblasts.

## 2. Materials and Methods

### 2.1. Antibodies, Inhibitors and Treatments

All antibodies, reagents, and inhibitors that were used in this study are listed in Appendix A. All inhibitors and drugs were dissolved in dimethyl sulfoxide (DMSO). After sub-culturing for 16~24 h, cells were treated with each inhibitor and the relevant drugs in fresh growth medium.

### 2.2. Isolation and Culture of Corneal Fibroblasts

Primary corneal fibroblasts were prepared from healthy corneas obtained from the eye bank of Yonsei University Severance Hospital. Culture and isolation of primary corneal fibroblasts were performed as previously described [33]. Donor confidentiality was maintained according to the Declaration of Helsinki. This study was approved by the Severance Hospital IRB Committee (4-2010-0013), Yonsei University. This study used wild-type (WT) (*n* = 4), heterozygous (HT) (*n* = 1), and homozygous (HO) (*n* = 3) human corneal fibroblasts immortalized by the expression of the catalytic subunit of human telomerase (hTERT) [36].

### 2.3. Cell Viability Assay

Cell viability after drug(s) treatment was determined by Presto Blue cell viability reagent (Invitrogen). The supporting information provides additional methodological detail (Appendix A).

### 2.4. Preparation of Cell Lysates and Western Blot Analysis

Cell lysates of the corneal fibroblasts were prepared in a radio-immunoprecipitation assay buffer (Cat # 89900, Thermo Scientific™ RIPA buffer) containing a protease inhibitor (Complete Mini Protease Inhibitor Tablet, Roche #1836170). Additional details are available in the Appendix A.

### 2.5. Mitochondrial Fractionation

Mitochondria were isolated from corneal fibroblasts using the mitochondria isolation kit for cultured cells (89874; Thermo Fisher Scientific Inc., Pierce, Rockford, IL, USA) according to the manufacturer’s instructions.

### 2.6. Quantification of mtDNA Copy Number and Deletion

Previously described methods were used for quantifying mtDNA copy number and deletion [37]. Briefly, one nuclear and three mitochondrial primer pairs were chosen [37]. Real-time PCR was performed in triplicate for each gene and the mean values of three measurements were used for statistical analyses. Additional details are available in the Appendix A.

### 2.7. Mitochondrial Network Imaging by Fluorescence and Electron Microscopy

The fluorescent dyes MitoTracker green FM (Thermo Fisher Scientific, M7514) and MitoTracker Red FM (Thermo Fisher Scientific, M22425) were used to monitor mitochondrial morphology in living cells according to the manufacturer’s instructions. Additional details are available in the Appendix A.

### 2.8. Statistical Analysis

Data are expressed as the mean ± SD. Significant differences (* *p* <  0.05, ** *p* < 0.01, *** *p* < 0.001) between the two groups were determined by Student’s *t*-test. When comparing more than two groups, significant differences were determined by one-way analysis of variance followed by Tukey’s multiple comparison test. All data were processed using the GraphPad Prism version 5.0 statistical package (GraphPad Software Inc, San Diego, CA, USA).

## 3. Results

### 3.1. Abnormal Mitochondrial Morphology in GCD2 Corneal Fibroblasts

We previously showed that mitochondrial morphology and dysfunction [34], as well as increased mitochondrial fragmentation [35] were exhibited by corneal tissues and cultured primary corneal fibroblasts obtained from a GCD2 patient, respectively. Hence, in this study, we hypothesized that mitochondrial dynamics, division, and fusion could be altered in GCD2 corneal fibroblasts. To address this hypothesis, we evaluated alterations in the mitochondrial network of GCD2 and WT corneal fibroblasts by EM and fluorescent microscopy. Ultrastructural analyses using EM revealed that GCD2 corneal fibroblasts had higher number of short, fragmented mitochondria with abnormal morphology (Figure 1B,D,F,G) compared to WT control (Figure 1A,C,E). Furthermore, GCD2 corneal fibroblasts had a higher number (2.21-fold) of total mitochondria compared to WT corneal fibroblasts (Figure 1H). Thus, as shown in Figure 1H, WT corneal fibroblasts had 28.69 ± 4.32% proximal cells with fragmented or short mitochondria that increased to 91.71 ± 9.82% in GCD2 corneal fibroblasts. In addition, WT corneal fibroblasts had 68.22 ± 4.63% proximal cells with long mitochondria that decreased to 7.50 ± 3.02% in GCD2 corneal fibroblasts (Figure 1H).

Next, we morphometrically assessed the mitochondrial network by staining live WT and GCD2 corneal fibroblasts with MitoTracker Red FM (Figure 1I–L) and MitoTracker Green FM (Figure 1M,N). In WT cells, mitochondria formed an interconnected network (Figure 1I, K,M). GCD2 HO corneal fibroblasts, however, showed small, punctuated, fragmented, and clustered mitochondria localized in the perinuclear region (Figure 1J,L,N). The GCD2 corneal fibroblasts also showed reduced tubular mitochondrial network compared to WT cells (Figure 1N). These data indicate abnormal mitochondrial fission/fusion dynamics in GCD2 corneal fibroblasts.

### 3.2. Altered Levels of Mitochondrial Dynamics-Related Proteins in GCD2 Corneal Fibroblasts

GCD2 corneal fibroblasts showed increased mitochondrial number and abnormal morphological features. To explore whether the alternation of mitochondrial dynamics underlies abnormal mitochondrial features, we assayed the status of six canonical dynamics-regulating proteins: OPA1, MFN1, MFN2, DRP1, phosphorylated DRP1 (pDRP1), and FIS1. Western blots showed that OPA1 and FIS1 levels were significantly reduced, but MFN1 and MFN2 levels were significantly increased in GCD2 corneal fibroblasts compared to WT cells (Figure 2A,B). Furthermore, DRP1 phosphorylation was significantly increased in GCD2 corneal fibroblasts compared to WT cells (Figure 2A,B). TGFBIp levels, however, were not significantly different between GCD2 and WT corneal fibroblasts (Figure 2A,B), indicating that TGFBIp levels may not be associated with mitochondrial abnormality.

### 3.3. OPA1 Reduction as the Major Factor in Mitochondrial Abnormalities in GCD2 Corneal Fibroblasts

Among the fission/fusion proteins, levels of OPA1—the inner mitochondrial membrane-localized protein involved in mitochondrial structure and dynamics—were decreased in GCD2 corneal fibroblasts; *OPA1* mutations cause dominant optic atrophy and optic nerve degeneration in retinal ganglion cells [10]. Hence, to explore the effects of OPA1 level reduction on cellular function, WT corneal fibroblasts were infected with lenti-shRNA targeting OPA1 (shOPA1). After 14 days, Western blotting revealed an effective knockdown (KD) (≈60%) of OPA1 expression (Figure 3A). The effect of OPA1 level reduction on mitochondrial morphology was analyzed by fluorescence microscopy. OPA1-KD WT corneal fibroblasts exhibited abnormal mitochondrial network and clustering (Figure 3B). Further, EM analyses demonstrated that reduction in OPA1 levels causes abnormal mitochondrial ultrastructure (Figure 3C, right and Appendix A). We also analyzed the effects of reduced OPA1 levels on the regulation of mitochondrial dynamic proteins; MFN1 level was constantly reduced, but DRP1 level was increased in OPA1-KD WT corneal fibroblasts (Figure 3D) and in mitochondrial fractions (Appendix A).

### 3.4. Reduction of OPA1 Level Increases the Susceptibility of Corneal Fibroblasts to Mitochondrial Stress-Induced Cell Death

OPA1 plays a key role in the mitochondrial regulation of apoptosis and controls apoptotic cristae remodeling independently from mitochondrial fusion [25]. Mitochondrial fission and cristae remodeling are proposed to facilitate the intracellular release of cytochrome c as the proapoptotic factor [38,39,40]. To determine whether reduction in OPA1 levels affects cell death and mitochondrial function, we used CCCP, an uncoupler of mitochondrial oxidative phosphorylation (OXPHOS). First, the cell proliferation assay showed that the cell viability of OPA1-KD corneal fibroblasts was significantly reduced from 6 h after 200 μM CCCP treatment compared to that of WT corneal fibroblasts (Figure 4A). Further, the apoptotic death rates were estimated by activating PARP1 and caspase-3 (casp-3). Addition of 100 μM CCCP increased cleaved (Cl)-PARP1 and Cl-casp-3 levels in OPA1-KD WT corneal fibroblasts (Figure 4B,C). Furthermore, addition of 200 μM CCCP enhanced the levels of these cleaved proteins in both WT (Figure 4B,C) and OPA1-KD WT corneal fibroblasts (Figure 4B,C). However, Cl-PARP1 and Cl-casp-3 levels significantly increased in 200 μM-treated WT corneal fibroblasts (Figure 4B,C) compared to 0~100 μM-treated WT and 0~50 μM-treated OPA1-KD corneal fibroblasts, respectively (Figure 4B,C). These apoptotic activations occurred in a time-dependent manner (Figure 4D,E).

### 3.5. Altered Levels of OXPHOS Complex Proteins and mtDNA Content in GCD2 Corneal Fibroblasts

To confirm mitochondrial dysfunction, we verified the levels of differential proteins by Western blotting. Five complex I–V electron transport chain proteins were analyzed. Western blotting showed that cytochrome-c oxidase subunit II (Complex IV subunit) and ATPase synthase subunit α (Complex V) levels were significantly reduced in GCD2 corneal fibroblasts (Figure 5A,B).

### 3.6. Mitochondrial Biogenesis Protein Levels Are Reduced in GCD2 Corneal Fibroblasts

In this study, complex IV and V levels were reduced in GCD2 corneal fibroblasts compared to WT cells (Figure 5), indicating that mitochondrial biogenesis may be reduced in these cells. As mitochondrial biogenesis is mainly regulated by the PGC-1α/NRF 1/TFAM signaling pathway, we determined the levels of mitochondrial mass proteins TOM20 (outer membrane) and TIM23 (inner membrane) [41]. TOM20 and TIM23 levels were significantly higher in GCD2 corneal fibroblasts (Figure 6A,B). Several mitochondria-related diseases are characterized by tissue-specific mtDNA instability such as mtDNA deletions and/or depletion. mtDNA deletions are characterized in dominant optic atrophy [42,43]. Hence, we screened GCD2 corneal fibroblasts for mtDNA deletions using PCR but did not detect any deletion (Appendix A). Subsequently, we quantified mtDNA contents using qPCR in GCD2 corneal fibroblasts and found that mtDNA amount was significantly decreased (Figure 6C). Western blotting showed that transcriptional regulators NRF-1 and NRF-2 act on the nuclear genes encoding constituent subunits of the OXPHOS system. NRF2 level was significantly reduced, but NRF1 level did not change significantly in GCD2 corneal fibroblasts (Figure 6D,F). Furthermore, NRF2 levels were significantly reduced in the nuclear fraction of GCD2 corneal fibroblasts (Figure 6E,G). The level of PGC1α protein, a master regulator of mitochondrial mass and function, also reduced significantly in the nucleus (Figure 6E,G), but its level was not different in the total cells extracts of GCD2 corneal fibroblasts (Figure 6D,F).

### 3.7. DRP1 Inhibitor Mdivi-1 Prevents Apoptosis Induced by Mitochondrial Stress

The phosphorylation of DRP1 (pDRP1) at Ser616 is known to stimulate fission [44]. Therefore, we investigated whether Mdivi-1(DRP1 inhibitor) can prevent apoptosis induced by mitochondrial stress in GCD2 corneal fibroblasts. Apoptotic rate was analyzed by PARP1 and casp-3 activation at 15 h after CCCP treatment (Figure 7A). Cleaved PARP1 and casp-3 (Cl-PARP1 and Cl-casp-3, respectively) levels between WT and GCD2 cells were similar in the treatment samples (lanes 1 and 5 in Figure 7). However, 200 µM CCCP treatment significantly enhanced Cl-PARP1 and Cl-casp-3 levels in both cell types but their levels were significantly higher in GCD2 corneal fibroblasts (Figure 7A–C). Further, when cells were co-treated with CCCP and Mdivi-1, Cl-PARP1 and Cl-casp-3 levels significantly reduced in cell types (lanes 4 and 8 in Figure 7A–C). These effects were more significant in GCD2 corneal fibroblasts than in WT cells (lanes 4 and 8 in Figure 7A–C), indicating that Mdivi-1 prevents apoptosis in GCD2 corneal fibroblasts induced by mitochondrial stress. In addition, only Mdivi-1 treatment enhanced PARP1 and casp-3 cleavage (lanes 2 and 6 in Figure 7A–C), but the difference between WT and HO corneal fibroblasts was not significant (lanes 2 and 6 in Figure 7A–C).

## 4. Discussion

For the first time, our study demonstrated that abnormalities in mitochondrial biogenesis and dynamics alter mitochondrial function in GCD2 corneal fibroblasts. Mitochondria can have defined intracellular distributions that can rapidly change according to physiological needs [12]. Mitochondrial length, size, and number change by constant division and fusion, thereby affecting their function [34,35]. Based on our observations, we suggest a possible treatment for abnormal mitochondrial dynamics in the corneal fibroblasts of GCD2 corneal dystrophy.

First, we found the profound alterations in mitochondrial dynamics lead to significant fragmentation of mitochondrial network. Mitochondria were significantly shorter in GCD2 corneal fibroblasts, which is consistent with previous work [35]. This abnormality could result from alternations in mitochondrial dynamics. We also investigated the balance between mitochondrial fission and fusion processes and found that the fusion and fission-related protein levels were different in GCD2 corneal fibroblasts. Among these proteins, OPA1 was notably downregulated, indicating that mitochondrial fusion was inhibited in GCD2 corneal fibroblasts. Considering that OPA1 controls mitochondrial dynamics, cristae integrity, energetics, mtDNA maintenance [16,22,45], and mitochondrial distribution [46], reduced OPA1 level may be the major factor in mitochondrial abnormalities in GCD2 corneal fibroblasts. This observation is supported by OPA1-KD in WT corneal fibroblasts that caused a similar abnormal mitochondrial distribution pattern, wherein such changes in mitochondrial morphology and distribution are comparable to those seen in GCD2 corneal fibroblasts. Furthermore, the effect of reduced OPA1 in mitochondrial network fragmentation in GCD2 corneal fibroblasts is strongly supported by a previous study reporting the KD of OPA1-induced mitochondrial fragmentation [47].

OPA1 is a dynamin-related protein belonging to the large GTPase superfamily, located in the inner mitochondrial membrane, and involved in mtDNA maintenance [22], indicating that mtDNA loss in GCD2 corneal fibroblasts may be likely due to reduced OPA1 level. Furthermore, OPA1 plays a role in maintaining mitochondrial cristae structure, and inner membrane integrity and potential, and preventing the release of cytochrome c (apoptotic trigger) [25], thus preventing apoptosis [48]. Hence, these data support the link between reduced OPA1 level and increased sensitivity to death in GCD2 cells, which is supported by previous studies reporting that reduced OPA1 levels increased mitochondrial fission and sensitivity to apoptotic stimuli [49]; conversely, its overexpression is anti-apoptotic [25,26].

Similar to our results, under pathological conditions, OPA1 has been well studied in dominantly inherited optic atrophy. The cells from patients with *OPA1* mutations had lower OPA1 levels and significant mitochondrial fragmentation compared with control cells [50]. Furthermore, these cells were profoundly depleted of mtDNA [50]. Mitochondrial fragmentation and mtDNA depletion were also increased in these cells due to *OPA1* mutations [50]. Moreover, in the absence of mitochondrial fusion, cells show a reduction in mtDNA amounts [51]. Collectively, these data indicate that reduced mtDNA content might be a result of decreased OPA1 level in GCD2 corneal fibroblasts.

Here we showed the possibility that in GCD2 corneal fibroblasts, an increase in MFN1/2 (outer membrane fusion protein), TOM20, and TIM23 levels could activate mitochondrial biogenesis, while a decrease in mtDNA contents, OPA1 (inner membrane fusion protein), FIS1 (outer membrane fusion protein), and complex IV and V levels could inhibit mitochondrial biogenesis. However, we cannot rule out the possibility that enhanced levels of TOM20 and TIM23 that import proteins into the mitochondria might result from an induced improvement in mitochondrial membrane potential in GCD2 corneal fibroblasts, requiring more studies to determine whether the mitochondrial membrane potential contributes to mitochondrial pathophysiology in GCD2 corneal fibroblasts.

We further extended this study by investigating the mitochondrial biogenesis, which is tightly regulated by the crosstalk between mitochondrial and nuclear genomes and is coordinated by PGC1α, NRF1, and NRF2 [52]. PGC1α co-activates NRF1 and NRF2, which in turn coordinate the expression of TFAM. Accordingly, the levels of these proteins appear to play an important role in the mitochondrial pathophysiology of several degenerative diseases. For example, PGC1α and NRF2 levels were reduced under pathological conditions and are associated with mitochondrial dysfunction and oxidative stress [31,52,53]. Similarly, PGC1α and NRF2 levels are reduced in the nuclear fraction of GCD2 corneal fibroblasts, indicating that mitochondrial biogenesis is downregulated in GCD2 corneal fibroblasts. However, the mechanism about the increase of number of fragmented mitochondria in GCD2 corneal fibroblasts is yet to be elucidated. We suggest two probabilities: first, defective mitophagy could lead to an increase in the fragmented mitochondria in GCD2 corneal fibroblasts. This notion can be supported by lysosomal dysfunction [54] and impaired autophagic flux in GCD2 corneal fibroblasts, respectively [35], suggesting the need to study the roles of mitophagy in mitochondrial dysfunction in GCD2. Second, an increase of Ser^616^ phosphorylation in GCD2 corneal fibroblasts might result in inhibition of DRP1 GTPase activity, preventing the protein from translocating from the cytosol to the sites of mitochondrial division, thereby inhibiting mitochondrial fission [55,56]. Thus, increased pDRP1 would cause increase of mitochondrial fission and result in accumulation of the fragmented mitochondria in GCD2 corneal fibroblasts.

DRP1-triggered excessive mitochondrial fragmentation contributes to apoptotic cell death under pathophysiological conditions and therefore it has emerged as a promising therapeutic target for aging and age-related disease [57]. Further, Mdivi-1 inhibits DRP1-dependent mitochondrial fission and exhibits protective action in several neurodegenerative disease models [57,58]. In our study, Mdivi-1 reduced CCCP-triggered apoptotic death in both WT and GCD2 corneal fibroblasts. However, it induced apoptosis in both WT and GCD2 corneal fibroblasts, making it unsuitable for treating GCD2 corneal dystrophy.

Corneal fibroblasts that produce procollagen and collagen reside between the collagen lamellae, and are responsible for secreting corneal ECM components including TGFBIp which is required for the maintenance of normal corneal structure and function [9,59]. Given that the pathogenic mutant-TGFBIp is accumulated in between the collagen lamellae of the GCD2 corneal stroma, maintaining normal mitochondrial dynamics and quality control in the GCD2 corneal fibroblast may help in delaying or preventing GCD2 corneal dystrophy.

## 5. Conclusions

In conclusion, our results indicate that the dynamic mechanisms regulating mitochondrial quality control are skewed in GCD2 corneal fibroblasts. Modulation of mitochondrial dynamics and biogenesis may indicate a new mitochondria-targeted strategy to attenuate GCD2 progression. Nevertheless, further studies are needed in the future to determine whether mitochondrial abnormalities directly result from mutant-TGFBIp, and to define more precisely the role of mitochondrial quality control including mitophagy in the pathophysiology of TGFBI-linked corneal dystrophy.

## Figures and Tables

**Figure 1 genes-14-00566-f001:**
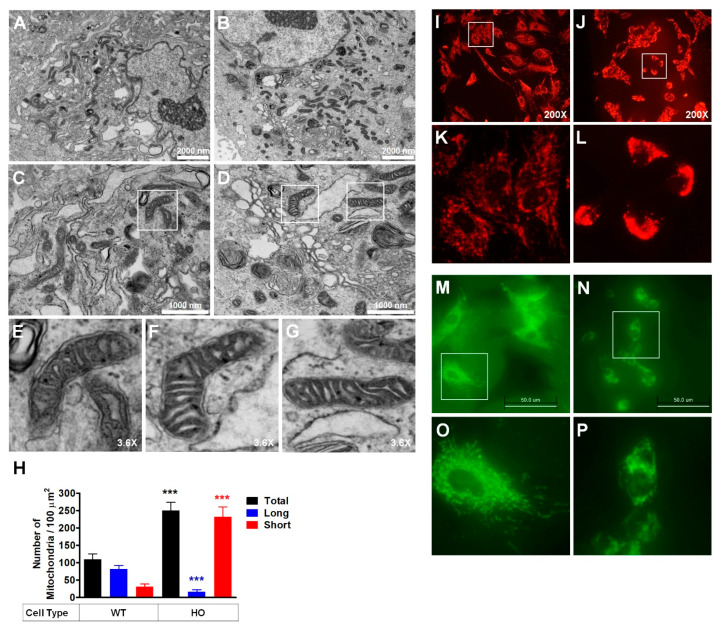
Altered mitochondrial morphology and network. (**A**–**G**) Representative images showing mitochondrial morphology in WT and GCD2 corneal fibroblasts. EM micrographs of a thin section of corneal fibroblasts show the structure of mitochondria in WT and GCD2 corneal fibroblasts. Many abnormal mitochondria (short or fragmented) were observed in GCD2 cells compared with WT cells. Scale bars are 2000 nm (**A**,**B**), and 1000 nm (**C**,**D**). (**H**) Graphs on the right show mitochondrial morphological features. Mitochondrial numbers were measured in 6 WT and 6 GCD2 cells. The ranges of the length parameters of the three shapes of mitochondria were arbitrarily determined as short (<500 nm), long (>500 nm), and total (short + long). The size of mitochondria was measured using Image (**J**) software (version 1.37, Wayne Rasband, NIH). Results represent the mean ± SD, *** *p* < 0.001. GCD2 corneal fibroblasts exhibit enhanced mitochondrial fragmentation. (**I**–**P**) Live imaging of mitotracker-stained mitochondria in WT and GCD2 HO corneal fibroblasts. Representative images of GCD2 corneal fibroblasts and control cells preincubated with MitoTracker Red/Green showing mitochondrial network formation. White boxes in upper panels (**I**,**J**,**M**,**N**) show corneal fibroblasts, which are magnified in lower panels (**K**,**L**,**O**,**P**). Scale bars are 50 μm (**M**,**N**).

**Figure 2 genes-14-00566-f002:**
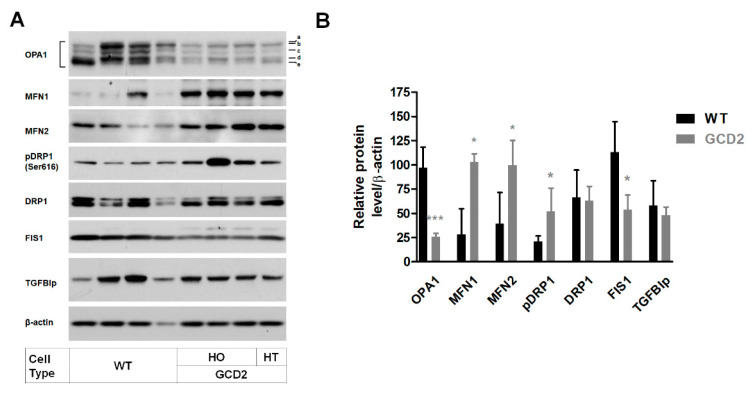
Mitochondrial fusion/fission-related protein levels. (**A**) A representative Western blot image for key mitochondrial dynamics proteins. OPA1 is regulated by complex patterns of proteolysis and alternative splicing. The OPA1 band patterns (a-, b-, c-, d-, and e-form) were expressed in WT and GCD2 corneal fibroblasts. (**B**) Graphs on the right show quantifications of different protein levels normalized to actin. OPA1 and FIS1 levels were significantly decreased in GCD2 corneal fibroblasts compared with those in control subjects, as were FIS1 protein levels. In contrast, MFN1, MFN2, and pDRP1 protein levels were significantly increased in GCD2 corneal fibroblasts compared with those in control cells. TGFBIp level was not significantly different between GCD2 and WT corneal fibroblasts. Error bars indicate mean ± SD. * *p* < 0.05 and *** *p* < 0.001.

**Figure 3 genes-14-00566-f003:**
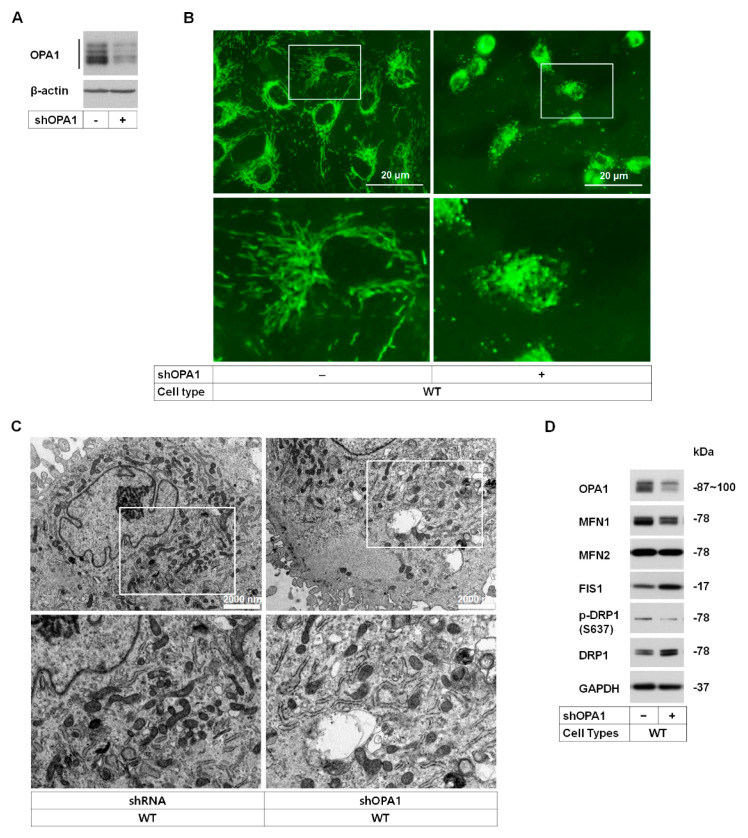
Knockdown of OPA1 leads to mitochondrial ultrastructural abnormalities and differential fusion/fission-related protein levels. (**A**) Expression levels of OPA1 in scrambled shRNA (-) and OPA1-specific shRNA-expressing (+) in corneal fibroblasts. β-actin was used as the loading control. (**B**) Confocal images of scrambled shRNA (−) and OPA1-specific shRNA-expressing (+) in WT corneal fibroblasts stained with MitoTracker Green. White boxes in upper panels show corneal fibroblasts, which are magnified in lower panels. Scale bars, 5 μm. (**C**) Representative EM images of mitochondrial ultra-structures of shRNA and shOPA1 in corneal fibroblasts. White boxes in upper panels show mitochondria of shRNA (left) and shOPA1 in WT corneal fibroblasts, which are magnified in lower panels. (**D**) Western blot analysis of the levels of fusion/fission-related proteins in OPA1-KD corneal fibroblasts.

**Figure 4 genes-14-00566-f004:**
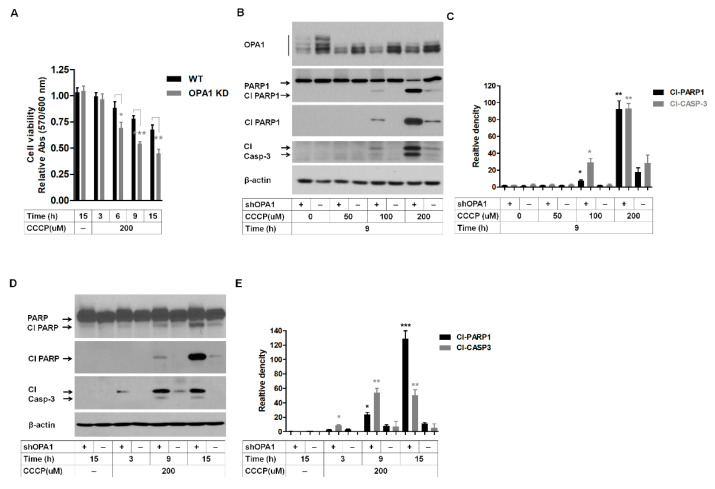
Reduction of OPA1 increases the susceptibility to cell death. (**A**) After treatment with CCCP (200 μM) for ~3 to 15 h, reversal of CCCP-induced toxicity either by 200 μM or 0 μM was investigated using Presto Blue cell viability reagent (Invitrogen). Experiments were performed in triplicate. Data were expressed as mean ± SD. (**B**) Cleavage of PARP1 and casp-3 were analyzed by Western blots following 9 h treatment with indicated concentrations of CCCP in corneal and OPA1-KD corneal fibroblasts. β-actin was used as a loading control. shOPA1; knockdown OPA1 cells with lenti-shRNA-OPA1, +; OPA1-KD cells, −; control cells, Cl; cleaved. (**C**) The quantification of cleavage bands of PARP1 and casp-3 in the previous panel (**B**). The bars represent the relative increase in treated versus untreated control and OPA1-KD corneal fibroblasts from 3 independent experiments ± SD. (**D**) Cells treated with 200 μM CCCP for 3, 9, and 15 h, and PARP1 and casp-3 cleavage was assessed in whole-cell extracts from cells treated as described above using Western blotting. β-actin was used as the loading control. (**E**) The quantification of cleavage bands of PARP1 and casp-3 in the previous panel (**D**). The bars shown are representative of the results of 3 independent experiments. * *p* < 0.05, ** *p* < 0.01, and *** *p* < 0.001 as compared to control, Student’s *t*-test.

**Figure 5 genes-14-00566-f005:**
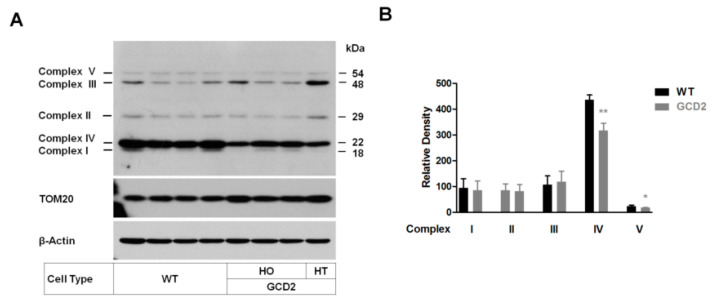
Levels of mitochondrial complex proteins. (**A**) Western blot analysis of the subunits of OXPHOS complexes I–V using the appropriate antibodies. Twenty-five micrograms of total cellular proteins from WT and GCD2 corneal fibroblasts were electrophoresed through SDS-PAGE gel and hybridized with an antibody cocktail specific for the subunits of each OXPHOS complex and with β-actin and TOM20 (mitochondrial membrane protein) as the loading control. A cocktail antibody comprising the following subunits of respiratory complex proteins was used: NADH dehydrogenase (ubiquinone) 1 β subcomplex 8 (NDUFB8; complex I), succinate dehydrogenase complex, subunit B, iron sulfur (SDHB/Ip; complex II), ubiquinol-cytochrome-c reductase core protein II (UQCR2; complex III), cytochrome-c oxidase subunit 2 (COXII; complex IV), and ATP synthase 5A (ATP 5A, Complex V). (**B**) Quantification of the levels of each of the abovementioned subunits were shown, respectively. The data are presented as % of proteins normalized to β-actin levels. Significances are shown as mean ± SD, *n* = 8, (* *p* < 0.05, ** *p* < 0.01).

**Figure 6 genes-14-00566-f006:**
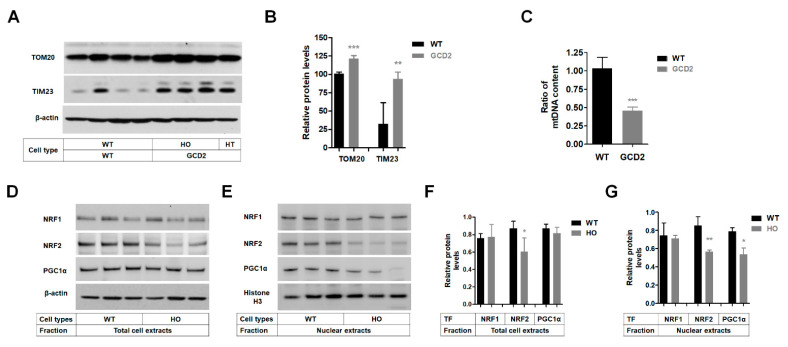
Mitochondrial biogenesis reduced in GCD2 corneal fibroblasts. (**A**) The levels of TOM20 and TIM23 were analyzed in WT and GCD2 HO corneal fibroblasts by Western blotting. β-actin was used as the loading control. Blots are representative of three independent experiments. (**B**) The quantification of TOM20 and TIM23 levels in WT and GCD2 HO corneal fibroblasts from three independent experiments ± SD. (**C**) Quantitative PCR Analysis of mtDNA content in WT and GCD2 corneal fibroblasts. mtDNA levels were significantly lower in GCD2 compared to WT corneal fibroblasts. The data are presented as mean ± SD from three independent experiments. (**D**) The levels of PGC-1α, NRF1, and NRF2 in WT and GCD2 HO corneal fibroblasts were determined by Western blotting in total extracts (**D**) and in nuclear fraction (**E**) of WT and GCD2 HO corneal fibroblasts. The quantification of PGC-1α, NRF1, and NRF2 protein bands in in total extracts (**F**) and in nuclear fraction (**G**) of WT and GCD2 HO corneal fibroblasts from three independent experiments. Histone H3 was used as an internal loading control. Data are presented as mean ± SD, *n* = 3 per group. (*n* = 3). * *p* < 0.05, ** *p* < 0.01, *** *p* < 0.001.

**Figure 7 genes-14-00566-f007:**
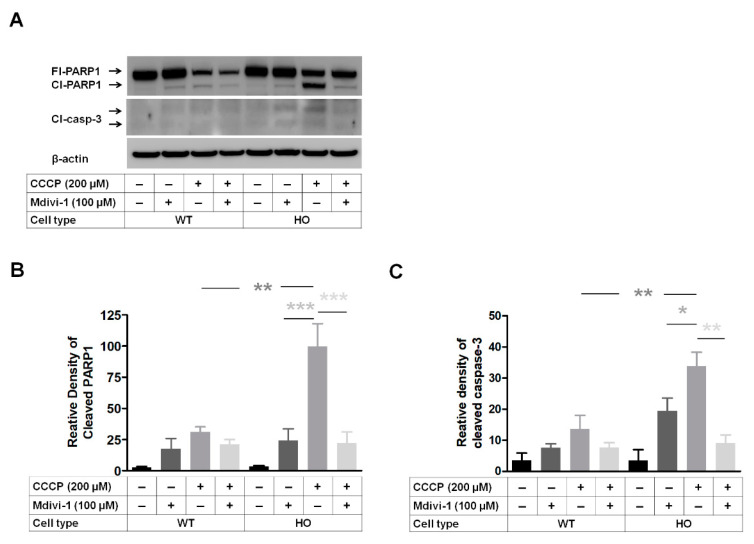
Mdivi-1 (DRP1 inhibitor) prevents CCCP-induced apoptosis of HO corneal fibroblasts. (**A**) WT and GCD2 HO corneal fibroblasts were treated with 100 μM Mdivi-1 in the presence or absence of 200 µM CCCP, and Cl-PARP1 and Cl-casp-3 were analyzed by Western blotting. (**B**) Optical densities of the PARP1 bands in (**A**). (**C**) The quantification of cleaved casp-3 levels in WT and GCD2 HO corneal fibroblasts. Western blotting of three independent experiments was performed. Bars represent mean ± SD, * *p* < 0.05, ** *p* < 0.01, *** *p* < 0.001 as compared to control, Student’s *t*-test.

## Data Availability

The original data presented in the study are included in the article. Further inquiries can be directed to the corresponding authors.

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
