# Peer review of "Reduced OPA1, Mitochondrial Fragmentation and Increased Susceptibility to Apoptosis in Granular Corneal Dystrophy Type 2 Corneal Fibroblasts"

_genes, 2023, doi:10.3390/genes14030566_

Round 1

Reviewer 1 Report

The authors describe reduced OPA1, mitochondrial fragmentation and increased  susceptibility to apoptosis in GCD2 derived fibroblasts. However, I have some major and minor comments, which should be addressed before considering the manuscript for publication.

Major comments:

1.      The authors write in Figure legend 2, that TOM20 and TIM23 levels are siginificantly increased in GCD2. However, there are no data at all for these two proteins in this figure.

2.      In Figure 2A, there seems to be four lanes of wt and four lanes of GCD2 (3 HO and 1 HT). Are these cell lines from different subjects? If so, please write this in the manuscript.

3.      Lines 176-177: The authors write in Figure legend: ‘OPA1 and FIS1 levels were significantly increased in GCD2 corneal fibroblasts compared with those in control subjects, as were FIS1 protein levels.’ However in the manuscript text and according to the Western blot and graph, OPA1 and FIS1 were significantly decreased in GCD2. The also write ‘as were FIS1 protein levels’. This is already written in the first half of the sentence.

4.      As the term HO is indicated in the cell lines used in this study under Material and Methods, the abbreviation HT is not explained at all. Does this refer to the heterozygous cell line? Please indicate.

5.      As GCD2 is an autosomal dominant disorder as stated in the introduction, why do the authors 3 homozygous cell lines and only one heterozygous cell-line? It seems more likely to me to collect three heterozygous cell lines than homozygous cell lines.  Also, the study of heterozygous cells would more reflect the in vivo situation in patients.

6.      The interpretation of Figure 3B is not obvious. Are there better pictures to explore the effect shOPA1 on the mitochondrial network? Can the picture be further improved by including DAPI stain?

7.      It would be appreciated, if the authors could mention and discuss further protein levels in section 3.3 concerning Figure 3D. E.g. FIS1 seems also to upregulated at protein level, which is contradictory to GCD2-cells.

8.      Is there a difference in GCD2 heterozygotes vs homozygotes in the reduction of Complex IV and Complex V subunits displayed in Figure 4?

9.      Title of Figure 7: Mdivi-1 (DRP1 inhibitor) prevents CCCP-induced apoptosis of OPA1-KD corneal fibroblasts. To my understanding, there are no OPA1-KD corneal fibroblasts used in this experiment. Is this title correct?

Minor comments:

1.      Line 33: The authors state that GCD2 is caused by the point mutation R124H in TGFBI. The term point mutation should be replaced by pathogenic variant and R124H (although probably historically correct) does not follow the actual sequence variant nomenclature (HGVS.org), which is p.(Arg124His) and should therefore also be adapted.

2.       Line 40: …’also known as corneal fibrobalsts in cultered in vitro.’ This end of the sentence is not really understandable. Could the authors rephrase this sentence?

3.      Lines 101-102: Could the authors please rephrase this sentence? ‘Supporting information provides more additional detail methods.’

4.      For lines 111-112: see point 3.

5.      Lines 116-117: see point 3.

6.      Figure 1: A scale bar in the Figures 1A-G would be highly appreciated. Although the authors state in the Figure legend: Bars: 2000 nm. It is not clears which bars are meant.

7.      Figure 1: The statement from the authors to observe that mitochondria are short or fragmented in GCD2 compared to wt is not obvious from Figure 1A-G. Could the author improve this?

8.      Figure 1H: Inclusion of the total mitochondrial mass per cell would be highly appreciated in this context. Does this change as well?

9.      Please indicate in Figure legend 1, which pictures refer to wt- and which pictures refer to GCD2-cells.

10.   Figure 2: Figure legend: ‘Graphs on the right’ should simply be replaced by (B). Otherwise (B) does not appear at all in the figure legend.

11.   Line 186: please add GCD2 to change the sentence into ‘were decreased in GCD2’

12.   Throughout the manuscript: Why is there a . after Figure? E.g. Figure.3C

13.   Line 244: (Complex IV subunit) instead of (Complex VI subunit) as there is no Complex VI in the OXPHOS-system.

Reviewer 2 Report

No suggestions. Well written and easily understood. 

Author Response

Thank you for reviewing our manuscript.

Reviewer 3 Report

In the manuscript by Choi et al. entitled “Reduced OPA1, Mitochondrial Fragmentation, and Increased 2 Susceptibility to Apoptosis in Granular Corneal Dystrophy 3 Type 2 Corneal Fibroblasts”, authors attempt to show the underlying mechanisms of mitochondrial dysfunction in granular corneal dystrophy type 2 fibroblasts. They found that these cells have structurally abnormal mitochondria with altered cristae morphology, enhanced mitochondrial fission, and fragmented mitochondrial tubular networks. Also, they revealed decreased levels of OPA1, which is essential for mitochondrial membrane fusion. These findings suggest that altered mitochondrial fission/fusion are molecular mechanisms that lead to mitochondrial dysfunction and, eventually, degeneration of GCD2 corneal fibroblasts.

Moreover, based on their findings, the authors suggest a possible treatment for abnormal mitochondrial dynamics in the corneal fibroblasts of GCD2 corneal dystrophy. However, later they state that this treatment is unsuitable for treating GCD2 corneal dystrophy since it triggers apoptosis in both WT and GCD2 corneal fibroblasts.

The findings are interesting; however, there are several points to be tackled:

Page 3, lines 133-141

The authors should define normal, long, and short-length mitochondrion. What are the sizes that they use to categorize the mitochondria?

 Page 5, line 192-194

On figure 3C, the ultrastructure of mitochondria is not visible. A higher magnification image should be included so the ultrastructure can be seen.

 Page 6, line 214

It should be 200 μM and not 200 μm.

 Page 10, lines 326-327

To make a statement that mitochondria were significantly shorter, they need to be measured, and the measurements need to be compared using statistics.

It cannot be stated that one is long and the other is short without clearly defining the ranges of the length parameters of all three phenotypes: short: xx μm, long: yy μm, normal: xy μm.

 Figure 1:

-Scales are not visible or are missing on images A-G.

- In (H), there is no explanation of what ***p means. Information on the statistical test used should be included.

- In the legend or on the panels A-G, it needs to be specified what images come from WT and GCD2 corneal fibroblasts.

- in the legend for I-N, it should be specified what cells are WT and GCD2 and labeled with Mitotracker red or green

- scale or magnification factor missing in M-N

 Figure 2:

- Incorrect information on OPA1 and FIS1 change in GCD2 – “OPA1 and FIS1 levels were significantly increased in GCD2 corneal fibroblasts compared with those in control subjects, as were FIS1 protein levels”, while in the text, there is opposite information.

- There is confusing information on TOM20 and TIM23 levels. However, there is no respective data shown on figure 2.

Figure 3:

Clear visible scale bars are missing on images B-C

- On C, we cannot see the ultrastructure of mitochondria. There should be higher magnification image included with higher resolution.

Figure 4

The figure y-axis label should be Relative density and not dencity.

Figure 5

In the legend description, it should be “data are shown” and not “Significant are shown”

Figure 6

Line 286; In the legend description correct the sentence below to make it clear:

“PCR are performed by 3 independent experiments.”

Figure 7

The legend should indicate what statistical test was used.
